# Acceptability and feasibility pilot randomised controlled trial of medical skin camouflage for recovery of women prisoners with self-harm scarring (COVER): the study protocol

Heather Mitchell,[1,2] Kathryn M Abel,[1,2] Brendan James Dunlop,[1,2] Tammi Walker,[3] Sandeep Ranote,[4] Louise Robinson,[5,6] Fiona Edgar,[1,7] Tracy Millington,[1] Rachel Meacock,[8] Jenny Shaw,[9] Kerry Gutridge[1]

For numbered affiliations see end of article.

**Correspondence to**
Dr Kerry Gutridge;
kerry.gutridge@manchester.ac.uk

## ABSTRACT

**Introduction** Self-harm in prison is a major public health concern. Less than 5% of UK prisoners are women, but they carry out more than a fifth of prison self-harm. Scars resulting from self-harm can be traumatising and stigmatising, yet there has been little focus on recovery of women prisoners with self-harm scarring. Medical skin camouflage (MSC) clinics treat individuals with disfiguring skin conditions, with evidence of improved well-being, self-esteem and social interactions. Only one community study has piloted the use of MSC for self-harm scarring.

**Methods and analysis** We describe an acceptability and feasibility pilot randomised controlled trial; the first to examine MSC for women prisoners who self-harm. We aim to randomise 20–25 women prisoners to a 6-week MSC intervention and 20–25 to a waitlist control (to receive the MSC after the study period). We aim to train at least 6–10 long-term prisoners with personal experience of self-harm to deliver the intervention. Before and after intervention, we will pilot collection of women-centred outcomes, including quality of life, well-being and self-esteem. We will pilot collection of self-harm incidents during the intervention, resources used to manage/treat self-harm and follow-up of women at 12 weeks from baseline. Data on recruitment, retention and dropout will be recorded. We aim for the acceptability of the intervention to prison staff and women prisoners to be explored in qualitative interviews and focus groups.

**Ethics and dissemination** Ethical approval for COVER has been granted by the North East–York Research Ethics Committee (REC) for phases 1 and 2 (reference: 16/NE/0030) and West of Scotland REC 3 for phases 3 and 4 (reference: 16/WS/0155). Informed consent will be the primary consideration; it will be made clear that participation will have no effect on life in prison or eligibility for parole. Due to the nature of the study, disclosures of serious self-harm may need to be reported to prison officials. We aim for findings to be disseminated via events at the study prison, presentations at national/international conferences, journal publications, prison governor meetings and university/National Health Service trust communications.

## Strengths and limitations of this study

► COVER is the first pilot randomised controlled trial of the use of medical skin camouflage for women who self-harm in prison.
► The study has been codesigned with experts-by-experience to test the delivery of a peer-led intervention.
► As a pilot, the sample size for the study is small, however, the research is designed to gather data on the feasibility and acceptability of delivering the intervention in prison rather than the efficacy of the intervention.
► The study will take place in one prison within the women's estate.

**Trial registration number** NCT02638974; Pre-results.

## INTRODUCTION
### Self-harm in women's prisons

Self-harm is defined as 'intentional self-poisoning or injury, irrespective of the apparent purpose of the act.'[1] The most common methods for self-harm in women's prisons are cutting and scratching followed by self-strangulation.[2] This complex behaviour is an increasing public health concern, not least because of its association with acute psychological distress and increased suicide risk.[2 3]

Self-harm is extremely prevalent and increasing in UK prisons. In the 12 months to December 2016 there were 7657 incidents of self-harm in female prisons, an increase of 4% on the previous year.[4] This is a rate of 1987 self-harm incidents per 1000 prisoners. Although women make up approximately 5% of the UK prison population, they are responsible for around a fifth of all prison self-harm.[4]

Research has shown that living with disfigurement from non-self-harm causes can have long-term physical and psychosocial effects, including reduced social interaction, increased social anxiety and reduced quality of life.[5][6] Furthermore, living with scars can be challenging in a society which values physical attractiveness.[7][8] It is likely that women prisoners with self-harm scarring experience similar psychosocial difficulties, for example, low self-esteem and interpersonal problems. These may be exacerbated by guilt and shame that women may feel because of their self-inflicted injuries.[9] There are, however, individuals who feel ambivalent about their self-harm scars, and while they may attempt to conceal scars in certain contexts, some feel confident and comfortable with their physical appearance.[10]

## Medical skin camouflage

Medical skin camouflage (MSC) uses British National Formulary-listed preparations to reduce the visibility of scarring or disfigurement,[11] with the potential to restore self-esteem, and aid recovery.[12–14] Products include skin-matched creams and powders that are waterproof and opaque and allow adherence to textured skin. All the products are 'borderline prescription' products that are available on National Health Service (NHS) prescription at each prescriber's discretion. A systematic review of the use of MSC in prisons yielded no available studies. Only a handful of published studies have evaluated the emotional/psychological benefits of MSC and all were in dermatological diseases or burns scarring. They report significant psychological benefit, improved social and sexual relationships and improved employability.[8][15][16] Despite these potential benefits, few services offer MSC for self-harm scarring.[17]

There has been little focus on how prisoners feel about their self-harm scars and no formally evaluated interventions to help women cope with any related psychosocial difficulties. This is the first study to formally deliver and evaluate an MSC intervention in a women's prison. Potential benefits of the intervention may include (1) increased self-esteem, confidence and quality of life; (2) empowering women to take part in work and social activities they might otherwise avoid; and (3) enhancing the strategies and interventions that prison staff have to work with self-harm.[18] Previous work by the research team has shown that there is a difficult relationship between prison staff and prisoners who self-harm and that staff feel restricted in how to help women.[19] This intervention may help staff to support women with self-harm scars and promote positive staff attitudes about self-harm and its management.

This study has been developed in collaboration with staff from North West Boroughs Healthcare NHS Foundation Trust (NWBP) who recently piloted an innovative camouflage service for service users with self-harm scars.[20] The 6-month pilot found that 95% of young people who used the MSC experienced improved confidence and ability to engage in activities.[20] To our knowledge, this is the first time that MSC has been evaluated in a mental health service and provided as part of a recovery package. The NWBP MSC service continues to be run in partnership with Changing Faces, a registered charity that uses volunteers to teach the MSC techniques to people in the community. This feasibility and acceptability study would provide insight into any benefits of using MSC in women's prisons and also any downsides, risks or unintended consequences.

## Phases 1 and 2

The MSC intervention and protocol used in our study were informed by the Changing Faces MSC training materials[21] and modified in phases 1 and 2 of the project. Phase 1 involved one focus group, with women prisoners with experience of self-harm (n=10) and one with prison staff (n=10). Both groups were conducted in Safer Custody meeting rooms and lasted between 60 and 90 min. The staff focus group explored and refined practical aspects of delivering MSC in the prison, including details of how participants would be recruited, where MSC clinics would be held and whether any MSC items would be unsuitable for prison use. The focus group with women prisoners helped to select the set of women-centred outcome measures and discussed their thoughts on long-term prisoners delivering the intervention. Women said they would prefer to be trained by other prisoners, particularly other women who have self-harmed. The rationale for recruiting long-term prisoners to deliver the intervention was to improve the sustainability of the intervention since they are likely to remain in the prison for a long time and can therefore continue training women to use MSC. Women also discussed the idea of completing a weekly diary; they thought this would be a good way of recording any thoughts or incidents of self-harm and some women had used a diary previously. Phase 2 involved adapting the MSC treatment intervention based on these focus groups, and producing the training and intervention protocols. The full analysis of the focus groups will be reported in a separate paper.

### Study aims

1. To evaluate the feasibility and acceptability of a randomised controlled trial (RCT) of MSC for women prisoners with self-harm scarring.
2. To assess the feasibility and acceptability of long-term prisoners delivering the MSC intervention.
3. To test the feasibility and acceptability of collecting a set of women-centred outcome measures before and after intervention, as well as a weekly self-harm diary.
4. To pilot follow-up of women at 12 weeks after baseline.
5. To test the feasibility and acceptability of collecting resource use data relating to self-harm incidents.

## METHODS AND ANALYSIS
### Design

This study is a feasibility pilot of an RCT, incorporating a qualitative component to assess the acceptability of MSC to women prisoners and prison staff. The study is taking

place in one UK closed women's prison. The research is funded by the National Institute for Health Research (NIHR) Research for Patient Benefit Programme (PB-PG-1013–32075). It was approved by North East–York Research Ethics Committee (REC) for phases 1 and 2 (reference: 16/NE/0030) and West of Scotland REC for phases 3 and 4 (reference: 16/WS/0155). The current protocol version is Version 6 (17/05/2017).

## Patient and public involvement

At the development phase of the research, a patient and public involvement group was conducted in one women's prison using a Patient and Public Involvement bursary from the NIHR Research Design Service North West. During this group women from the prison, who had self-harm scarring, contributed towards the research topic development through discussion of the possible impact of MSC. This informed the outcome measures for the research and the topic guides/interview schedules for the qualitative work. In phase 1 of the research, women in prison with self-harm scarring helped refine the design of the research assessing the burden of involvement in an RCT.

Two experts-by-experience joined the research team at the start of the research and contributed towards the design of all the materials for participants. In phase 3, one of these experts-by-experience will help train the long-term prisoners to be MSC practitioners having agreed to allow the women to practice application of the MSC on their self-harm scars. Another of our experts-by-experience, who is a trained qualitative researcher, analysed the phase 1 focus group data and will cofacilitate the staff focus group at the end of the research. A current prisoner, who works in Safer Custody, has agreed to help organise the participants' appointments in the prison and will sit on the project steering group. At the end of the research, our experts-by-experience will help us design a dissemination event for the women in prison that will involve presentations on the research outcomes. A plain English summary of the research will also be provided to women in the prison. We will also disseminate the research on the closed prison radio system.

## Sample size

Over 6 months (January 2017 to May 2017), we aim to recruit at least 6–10 long-term women prisoners to be trained in MSC. These women will then deliver the intervention to trial participants. The long-term prisoners will not be participants in the RCT, but will instead form an integral part of the research team delivering the intervention to the RCT participants.

Over 17 months (January 2017 to May 2018) we aim to recruit and consent 40–50 women prisoners to be randomised to receive either MSC or 'treatment as usual' in a waitlist control (to receive the MSC after the study period) design. Based on previous research[22] and recent figures from the study prison, we estimate that there will be around five to six eligible women per month. The sample size is based on a prediction that approximately half of these eligible women will be interested in the research.

## Participants and recruitment procedures

Recruitment procedures and advertisement strategies have been informed by the phase 1 focus groups. The research team will advertise the research at Safer Custody meetings attended by women prisoners. Leaflets and posters will be distributed to our local collaborators in the prison for display in different locations around the prison. Women will inform the local collaborators if they are interested in participating. We have one member of Safer Custody staff collaborating with the researchers and a woman prisoner from the Safer Custody team organising the research appointments. Prison staff will assess all volunteers to determine whether they would pose a risk to the researchers or other participants. Staff will also check the woman's sentence length to ensure she has enough time remaining on her sentence to take part in the study. In addition, healthcare staff will review the list of women to assess whether there are any health reasons why they might not be safe to participate. This has been usual practice across our decade of prisoner participation in research.

Vetted/screened women will be provided with an information sheet and offered the opportunity for the research team to visit, read through the sheet and answer any questions. Consent for the research will be agreed at least 24 hours after the information sheet has been read.

## Inclusion/exclusion criteria
### Phase 3
We aim to recruit 6–10 long-term prisoners with at least 10 years or more left on their sentence and who have experience of self-harm.

Discussions with prison staff suggested that the most suitable long-term women would be those who already hold a position of responsibility in the prison, for example, a peer supporter or trained Samaritan listener.

### Phase 4
We aim to recruit 40–50 women prisoners screened for date of release, with sufficient time left on their sentence to complete the intervention period. The women will have self-harm scarring anywhere on their body that they are happy to show to others, with at least some closed wounds (to allow the MSC to be applied).

All participants (phases 3 and 4) will be aged 18 or older and able to give written informed consent. Capacity to consent will be assessed by the experienced researchers (HM and KG) in collaboration with Safer Custody and Mental Health Care contacts in the prison. Participants will be excluded from the study if they are unable to provide written informed consent, or if they pose a risk to researchers (as assessed by the prison).

## Randomisation

In phase 4 internet randomisation (using an internet-based programme to randomise participants; www.sealedenvelope.com) will be carried out by the non-blind members of the research team, KG or HM, to allocate eligible women to MSC or waitlist control. Women in the waitlist group would receive one skin-matched prescription of MSC at the end of the research. Waitlist control has been chosen as the comparator to give all participants an opportunity to use the MSC. Participants randomised to the waitlist control would be aware of their allocation (figure 1).

## The intervention: MSC for self-harm scarring (adapted for delivery in a women's prison)

### Development

Outcomes of the phase 1 focus groups informed the development of the MSC intervention materials. In addition, two service user researchers (FE and TM) provided guidance to the research team, focusing on whether the intervention materials were suitable in terms of readability and sensitivity.

The intervention package consists of the training manual and four additional documents. The main training manual has been adapted from training manuals used by Changing Faces.[21] The adapted materials have been reviewed by a representative from the charity, to ensure that all key learning and safety points are covered.

### Manual content

The 34-page training manual has 13 sections that are listed and briefly described in table 1.

### Accompanying documents

1. A single sheet of key learning points for long-term prisoners covering safety issues such as how to protect trial participants, for example, breaking confidentiality if a woman discloses something which suggests she or someone else is at risk of harm.
2. A monitoring sheet for long-term prisoners to be used in weekly meetings with the research team. The form will help identify whether any further training or support is required.
3. An appointment checklist for long-term prisoners breaking down the 14 core steps in an MSC appointment, from laying out the kit, to completing a prescription record card.
4. A DO's and DON'Ts sheet for trial participants: this covers reminder points, including those related to safety and hygiene (eg, always keep lids on the products) and some rules relating to continued participation in the trial (eg, don't trade or share the products as only one prescription will be provided, added at the request of prison staff).

### Delivery of the intervention

Three stages of delivery: (1) training sessions for long-term prisoners, (2) skin camouflage clinics run by long-term prisoners for trial participants, (3) prescription of MSC products by prison healthcare.

1. Members of the research team aim to deliver a half-day group training session to 6–10 long-term women prisoners. During this session, the research team will work through the training manual, answering any questions and giving practical demonstrations of colour matching, application techniques and powdering. Participants will participate in practical activities to ensure that they have understood the training and are competent in MSC. There is scope for the training time to be extended if the women require more practice.
2. The aim is that regular skin camouflage appointments will be run by the trained long-term prisoners. The appointments will be held during the core prison day and will not interfere with the women's income. All participants will be seen individually for 1 hour; the intervention group will be seen as soon as possible after they have been randomised and the waitlist control group will be seen after they have completed their 12-week follow-up. During this appointment, the long-term prisoner will provide the woman with information about the MSC creams and powders; including allergy checks to ensure the woman can safely use the products. The long-term prisoner will then perform a colour match for the participant and demonstrate the application techniques. The participant will then practise applying the camouflage creams themselves until they are happy with the results. The long-term prisoner will then complete a record form to be given to healthcare.
3. The aim is for a nurse prescriber from healthcare to meet with all participants (the intervention group at the start and the waitlist control group at the end of the research) and write a prescription for 1× camouflage cream and 1× camouflage powder. Women will be informed that they will only be given one prescription for the duration of the study. The amount of camouflage cream required will depend on the extent of the participant's scarring, but based on the NWBP pilot[20] we anticipate that one prescription will be enough for a 3-month period.

Continued provision of the MSC products after trial is not envisaged at this stage within the study prison. However, all participants will be given a letter that they have the option to give to their general practitioner in the community that will detail their MSC prescription, and will recommend that the product is prescribed to them.

### Assessing feasibility and acceptability

We will assess the feasibility of recruiting and randomising women to MSC versus waitlist and of long-term prisoners delivering MSC appointments. We will examine use of the MSC, attrition (number of dropouts at each time point) and retention (the proportion of participants who complete the intervention period). The feasibility of delivery in a prison setting (ie, location, duration of

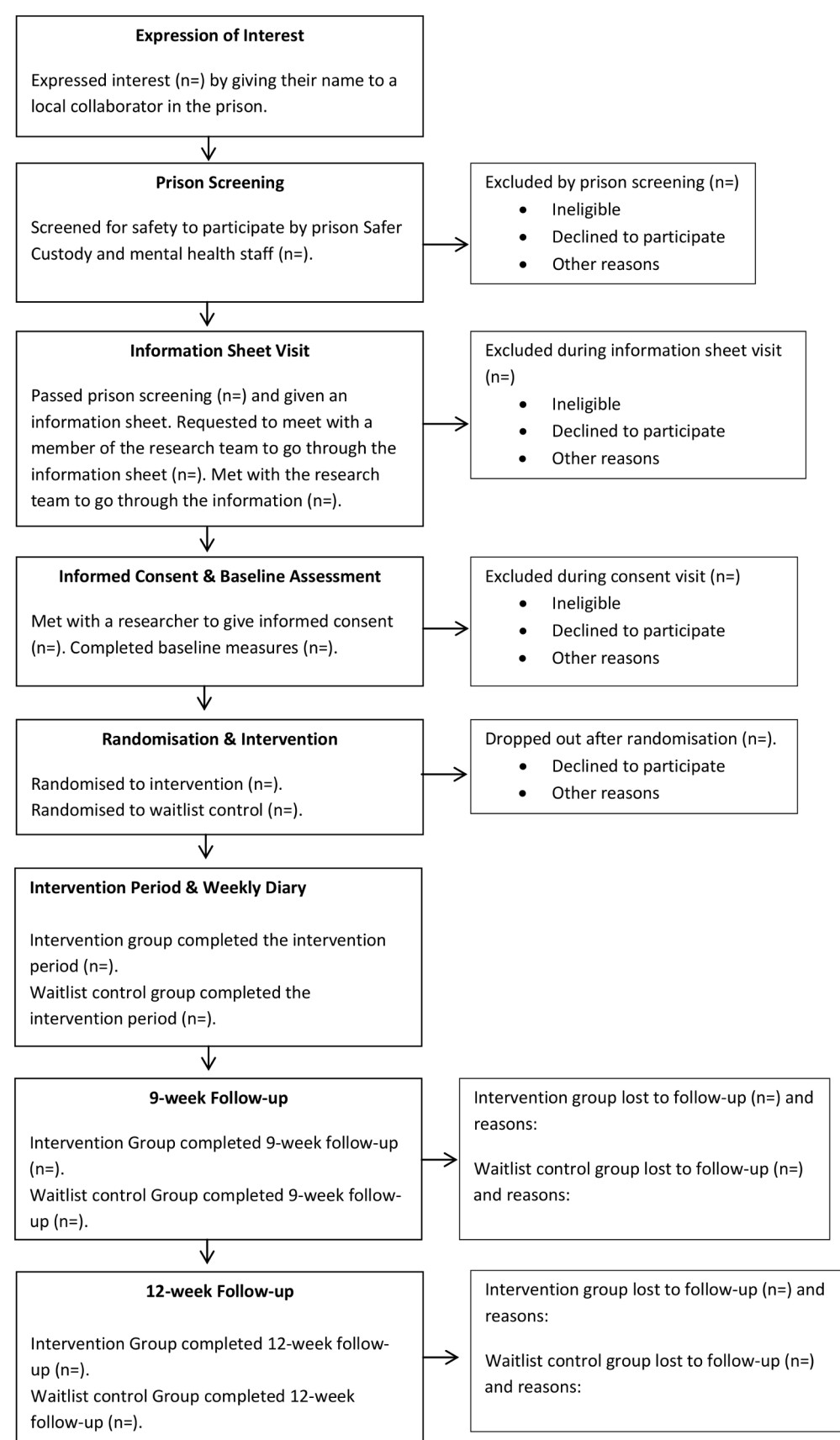

**Figure 1** Consolidated Standards of Reporting Trials (CONSORT) diagram.

**Table 1** Sections of the training manual

| Section No | Section name | Overview | Key learning point |
|---|---|---|---|
| 1 | Self-harm | This section aims to help women understand the different forms that self-harm can take and different reasons why women self-harm. | Different people have very different reasons for self-harming and it is therefore important to not make assumptions. |
| 2 | Working with women who self-harm in the COVER project | This section covers how to manage confidentiality and how to work with women who self-harm, for example, being respectful, don't judge the participant, the limits of confidentiality. | To manage and understand the limits of confidentiality, for example, if she discloses something that puts her or someone else at risk, and what to do if a woman becomes upset. |
| 3 | Hygiene | This section covers how to run a hygienic skin camouflage clinic and how to keep the kit clean. | Hygiene rules to follow during an appointment. |
| 4 | Communication | This section covers communication rules, including how to manage participant expectations, for example, setting realistic expectations for what MSC can achieve. | Understanding the importance of helping the clients to express their wishes and working with them to achieve the best results. |
| 5 | The skin and skin types | An overview of preparing the skin for application of MSC and how to ensure safe usage, for example, by checking for allergies. | How to prepare the skin and when it is not safe to use the products. |
| 6 | Overview of the kit | This section describes the items in the MSC kits and how to lay them out in a logical order. | Laying the kit out in a logical order will help the practitioner to quickly identify the products. |
| 7 | Colour matching | This section covers colour matching. This will involve some practical activities on identifying colour tones and colour matching. | To be able to identify tones in the creams and perform a colour match. |
| 8 | Brush technique | An overview of the brush technique and when/how to use it. | To understand when and how to use brushes. |
| 9 | Finger technique | An overview of the finger technique and when/how to use it. | To understand when and how to use the finger technique. |
| 10 | Sponge technique | An overview of the sponge technique and when/how to use it. | To understand when and how to use sponges. |
| 11 | Spreading technique | An overview of the spreading technique and when/how to use it. | To understand when and how to use the spreading technique. |
| 12 | Working with powder | An overview of how to use powder to set the MSC creams. | To understand the purpose of powder, and how to apply it. |
| 13 | Completing the record card | This section covers how to complete the participant record card, including what to do with the record card after the appointment. | What to include on the record card. |

MSC, medical skin camouflage.

training, peer delivery) and the acceptability of the intervention to women and staff will be assessed using qualitative interviews and focus groups. The feasibility of undertaking a full-scale RCT of MSC for women in prison will be assessed by studying recruitment (the proportion of eligible participants consenting to join the study) and completeness of outcome measures at baseline, after intervention (approximately 9 weeks from baseline to include the time taken to receive the MSC and 6 weeks of MSC use) and at follow-up (12 weeks from baseline). We have included a 12-week follow-up to assess retention and attrition over a longer period of time. Data will be collected on reasons for ineligibility, non-consent and dropout, including when the participant dropped out/withdrew from the study.

### Outcome measures for future RCT design

The aim is that all participants in both groups (MSC and waitlist control) will be asked to complete a set of quantitative outcome measures at baseline (0 week), after intervention (approximately 9 weeks later) and at follow-up (approximately 12 weeks from baseline). This will help us assess the feasibility and acceptability of these measures for a future clinical and cost-effectiveness RCT. Outcome

measures will be administered by the project manager (PM), trained research assistant (RA) or research nurses from the NIHR clinical research network. The PM and the RA will be unblinded to the randomisation outcome and will therefore only administer baseline measures; administration of measures at any other time point by these individuals may bias results. The research nurses will be blinded and will complete the postintervention and 12-week follow-up assessments. All research assessments (which we anticipate will last approximately 1 hour) will take place in a private room in Safer Custody. The PM, RA or research nurse will complete a case report form for each participant; recording any additional notes on each participant, for example, reasons for questionnaire non-completion. Given the sensitive nature of some of the selected outcome measures, we have consulted with women, Safer Custody staff and healthcare/mental health staff to develop procedures to protect and support participants. If, at any point during a research assessment, the woman becomes agitated or distressed, we will ask them if they would like to take a break or if they want to resume the assessment on another day. If the researcher has any concerns for the woman, they will alert the local collaborator who will ensure it is dealt with accordingly using existing prison support systems. The participant information sheet outlines that the researcher is obligated to inform the prison if there is a risk to the participant's health, safety or well-being. For this study, this will include reporting high suicidal ideation and high risk of serious self-harm.

We aim to administer a selection of outcome measures (see table 2) to all participants at baseline, 9 weeks and 12 weeks after baseline. Two of these measures, the Dermatology Quality of Life Index (DQLi)[23] and Rosenberg Self-Esteem Scale (RSES),[24] were added following focus group discussions on the psychological and interpersonal impact of scars. At baseline, we also aim to use a bespoke demographic and personal history questionnaire to collect relevant personal information including age, ethnicity, whether they are on remand or sentenced, past experience of contact psychiatric services, drug dependence and experiences of domestic violence, sexual abuse and parental neglect. We aim to collect this information to check whether the two randomised groups have similar backgrounds. With women's permission, our local collaborator or a research nurse will access information on key forensic and clinical characteristics from CNomis, SystemOne (the prison electronic medical records) and from Assessment Care in Custody and Teamwork documentation; these systems will be accessed by prison staff unless the researchers are granted access permission. Forensic characteristics will include types of offence (violent or non-violent), sentence length and stage of sentence, and clinical characteristics will include psychiatric diagnosis and history. We aim to administer the Deliberate Self-Harm Inventory[25] at baseline: a 17-item questionnaire that assesses the history and frequency of self-harming behaviours. We also aim to administer the Zanarini Rating Scale for Borderline Personality Disorder (ZAN-BPD) at baseline as a measure of borderline psychopathology.[26]

We aim to examine whether the Warwick-Edinburgh Mental Well-Being Scale (WEMWBS)[27] is a suitable primary outcome for a full-scale RCT. The WEMWBS is a 14-item scale of mental well-being covering subjective

| Table 2 | Participant assessment schedule | | | | | |
|---|---|---|---|---|---|---|
| | | | **Time point** | | | |
| | | Duration | | | | |
| **Assessment tool** | **Brief description** | **(min)** | **Baseline** | **Postintervention** | **3 months** | |
| Personal history questionnaire | Sociodemographic/life history | 5 | X | | | |
| DSHI | Methods/history of self-harm | 10 | X | | | |
| WEMWBS | Mental well-being | 5 | X | X | X | |
| BSS | Suicidal ideation | 10 | X | X | X | |
| BDI-II | Depression | 10 | X | X | X | |
| BHS | Hopelessness | 5 | X | X | X | |
| DQLi | Self-harm scarring quality of life | 5 | X | X | X | |
| RSES | Self-esteem | 5 | X | X | X | |
| Zanarini Rating Scale | Borderline personality disorder | 5 | X | | | |
| EQ-5D-5L | Generic health | 5 | X | X | X | |
| SF-12 | Generic health/quality of life | 5 | X | X | X | |
| Qualitative interview | Acceptability and feasibility | 30 | | | X | |
| Total time burden | | | 70 | 65 | 95 | |
| Self-harm diary | Self-harm thoughts and incidents | Weekly from baseline to 3 months | | | | |

BDI-II, Becks Depression Inventory II; BHS, Beck Hopelessness Scale; BSSI, Becks Scale for Suicidal Ideation; DQLi, Dermatology Quality of Life Index; DSHI, Deliberate Self-Harm Inventory; EQ-5D-5L, EuroQol Five-Dimensional Questionnaire, Five-Level Version; RSES, Rosenberg Self-Esteem Scale; SF-12, 12-Item Short-Form Health Survey; WEMWBS, Warwick-Edinburgh Mental Well-Being Scale; ZAN-BPD, Zanarini Rating Scale for Borderline Personality Disorder.

well-being and psychological functioning, in which all items are worded positively and address aspects of positive mental health. The WEMWBS has high internal consistency (α=0.91) and test–retest reliability (0.83).[27] This measure would be used to calculate study power in a full-scale subsequent trial.

Becks Scale for Suicidal Ideation (BSS)[28]: a 19-item instrument measuring intensity, duration and specificity of thoughts about committing suicide. The BSS has high internal consistency (0.89) and high inter-rater reliability (0.83).[28] The BSS has been successfully used in a pilot trial of Psychodynamic Interpersonal Therapy for women prisoners who self-harm.[22]

Becks Depression Inventory (BDI-II)[29]: a 21-item scale measuring symptoms of depression. The BDI-II has high internal consistency and a test-retest reliability ranging from 0.73 to 0.96.[30]

Beck Hopelessness Scale (BHS)[31]: a 20-item self-report inventory designed to measure three major aspects of hopelessness: feelings about future, loss of motivation and expectations. The BHS has high concurrent validity (0.86) and high reliability (α=0.91).[31]

Prison-adapted DQLi[23]: a 7-item questionnaire adapted from a validated 10-item scale that has been used in over 40 different skin conditions in over 80 countries. Test–retest reliability has been found to be high (0.99).[23]

RSES[24]: a 10-item Likert scale with items answered on a 4-point scale—from strongly agree to strongly disagree. The scale measures self-esteem and has been used in prison research.[32] Internal consistency ranges from 0.77 to 0.88 and test–retest reliability ranges from 0.82 to 0.85.[24]

EuroQol Five-Dimensional Questionnaire, Five-Level Version (EQ-5D-5L)[33]: a generic preference-based measure covering five domains of health-related quality of life (mobility, self-care, usual activities, pain/discomfort, anxiety/depression). Test–retest reliability is high and ranges from 0.78 to 0.87, with convergent validity at 0.64.[34]

12-Item Short-Form Health Survey (SF-12) is a shortened version of the SF-36,[35] consisting of 12 questions covering eight dimensions of health: physical functioning, role limitations—physical, bodily pain, general health, vitality, social functioning, role limitations—emotional, and mental health. Test–retest reliability ranges from 0.76 to 0.89 and relative validity ranges from 0.43 to 0.93.[34]

To reduce attrition, we aim to seek consent at baseline for women who have been transferred or have left prison during the study period to be followed up in person at other prisons or in a public place in the community, following a lone worker policy.

In addition to the outcome measures listed above, we also aim to ask trial participants to complete a weekly diary every week from their baseline assessment. Prison staff and women prisoners in the phase 1 focus groups proposed the use of a weekly diary; some of the women had completed a diary of self-harm thoughts and events in the past and found it helpful. The research team will collect the diary each week. The diary will ask questions about any thoughts or acts of self-harm that have occurred during the week and any life events that have impacted on their self-harm during the week. Women will also have a free-text space to add additional comments.

We also aim to pilot the collection of resource use data so that we can determine if it is feasible to gather these data in a larger trial, with a view to calculating the cost of treatment in comparison to usual care. This will be collected using the Secure Facilities Service Use Schedule[36] and a bespoke resource use questionnaire. Resource use data are likely to be extracted by the local collaborator from systems such as CNomis and Officers logs. Prison staff will redact any confidential information. We also aim to use these systems, together with SystmOne, to extract data on self-harm incidents that occurred during the intervention. If we successfully extract the data we will then triangulate prison records of self-harm incidents with women's self-reported incidents. We will record the time taken by prison staff and healthcare staff to extract this information.

To inform a future cost analysis, we also aim to record the time spent by Changing Faces training the researchers in MSC, time spent by the research team training long-term prisoners to become skin camouflage practitioners, time spent by long-term prisoners delivering the intervention and quantities of MSC products prescribed.

### Qualitative data

We aim to conduct interviews with all women in the MSC group (n=20) at the end of the study, to assess the acceptability of the intervention to service users. The interviews with women will explore their views on applying MSC, how long it stays on for, how useful they found it and any positive or negative effects on their everyday life, mood, self-esteem and self-confidence. The topic guides have been developed in consultation with two service user researchers and informed by outcomes of the phase 1 focus group.

We also aim to interview the long-term prisoners to assess their experiences of being an MSC practitioner, in terms of the acceptability of the training, mentoring/support from the research team and any benefits or difficulties working with participants.

In addition, we aim to conduct a focus group with prison staff from different disciplines (including Safer Custody staff, prison officers and healthcare staff) that have been in contact with women involved in the trial. The focus group would explore acceptability of the intervention from a staff perspective, including what they thought about prisoner delivery of the MSC intervention and whether the intervention has had a positive, or negative, impact on their job or their relationships with women prisoners. All interviews and the focus group will use semistructured topic guides with open-ended questions that should enable us to explore in-depth the aspects of the intervention that worked well, the aspects that did not work well and things that could be improved. With permission from participants,

interviews will be audio recorded. All recordings will then be transcribed verbatim and analysed using thematic analysis.[37]

We aim to assess fidelity to the MSC intervention by (A) observing the long-term prisoners at the end of training covering one of our service user researcher's scars, and (B) audio recording 10% of the training sessions which will be rated for fidelity to the training manual by an independent researcher.

## Data analysis
### Quantitative analyses
We shall compare means before and after treatment using descriptive statistics, including SDs and CIs for outcome variables to inform sample size estimates for a future RCT. We will also present descriptive statistics on recruitment and retention of participants in both groups, including reasons for dropout at different stages.

We shall assess the feasibility and relevance of both the EQ-5D-5L and SF-12 for the prison population through correlation between changes from baseline to follow-up of these and other piloted measures (WEMWBS, BSS, BDI, BHS, RSES); and examination of completion rates. Descriptive analysis of health-related quality of life data will also inform the suitability of the measures for future clinical and economic evaluations of the intervention.

Resource use collection will also be assessed through time taken to complete questionnaires, completion rates and ability to obtain included resource use categories to inform suitability of resource use categories in a future economic evaluation. Descriptive analysis of resource use data will also inform future trial design.

### Qualitative data
Qualitative data will be analysed using thematic analysis[37]; analysis which will be conducted by the RA and PM and checked for accuracy by an independent researcher. Preliminary codes and categories are assigned to the text[38] and emergent themes subject to constant comparison and examined for goodness of fit until a final set of key themes identified.[39] Adopting an inductive, iterative approach, data analysis will commence with the first interview.

### Data entry and storage
Written consent forms and completed questionnaires will be removed straight to the University of Manchester. Participants will be given a unique participant number that will be used on questionnaires and the electronic database. A password-protected document will link participant names and numbers. Any identifying personal data (eg, consent forms) will be stored separately from other research data. In the University of Manchester this will mean storage in the locked limited access corridor. Electronic databases will be stored on an encrypted space on University of Manchester computers. The RA would enter all data and the PM will carry out 10% checks for accuracy.

## ETHICS AND DISSEMINATION
Ethical approval for COVER was granted by the North East–York REC for phases 1 and 2 (reference: 16/NE/0030) and West of Scotland REC 3 for phases 3 and 4 (reference: 16/WS/0155).

### Adverse events
All participants will be women who have a history of self-harm. Therefore, self-harm incidents are an expected event and not necessarily a serious adverse event. All adverse events, including incidents of self-harm, will be recorded and reported to the PM. In consultation with prison staff and the prisoner, the research team will assess the seriousness of the adverse event and whether it is related to project participation; events that are judged as serious and unrelated will be reported to the sponsor only. Events judged as serious and related to project participation will be reported to the research sponsor, host NHS trust and West of Scotland REC.

### Dissemination
We aim for our findings to be disseminated to prisoners, prison staff and to the wider stakeholder (academic and clinical) community via showcase events at the study prison, presentations at national and international conferences, journal publications, safer custody and prison governor meetings and university/NHS trust communications. During dissemination, we will hold discussions with key personnel from NHS England and Her Majesty's Prison and Probation Service (HMPPS) regarding future provision of the intervention.

## DISCUSSION
Despite the large number of women in prison who self-harm (or who have self-harmed in the past and are living with scarring), there are little/no evidence-based interventions which aim to improve self-esteem, confidence and well-being. This low-cost intervention has the potential to improve women's mood and how they feel about themselves.

Our phase 1 focus groups suggested that many women prisoners who repeat self-harm struggle on a regular basis with negative feelings about their scars, for example, they have to cover them in front of others/family for fear of being judged adversely or upsetting them; they are a constant reminder of bad times or they lack confidence in their bodies because of scars. A prisoner-delivered MSC intervention could reduce such distress women prisoners experience and help them reintegrate into the community without the additional burden of being judged because of their scars.

This intervention was implemented successfully in a community mental health service. We, therefore, anticipate that, with the support of prison staff and long-term prisoners, COVER will provide a beneficial resource to improve well-being in an often-neglected population.

Engaging long-term prisoners in the delivery of MSC clinics should increase the sustainability of the intervention if it were to be commissioned in future and provide meaningful work for women prisoners, offering a valuable opportunity to improve relationships between prisoners and contribute towards a therapeutic community with the prison. Peer support schemes, such as the Samaritan's Listener scheme which runs across many UK prisons, are increasingly popular, enabling prisoners to develop a range of transferable skills and reducing the burden of distress and self-harm management for prison staff. If successfully implemented, COVER will run alongside these peer support services and provide additional help for women who self-harm.

**Author affiliations**
[1]Centre for Women's Mental Health, Division of Psychology and Mental Health, School of Health Sciences, Faculty of Biology, Medicine and Health, University of Manchester, Manchester Academic Health Science Centre, Manchester, UK
[2]Greater Manchester Mental Health NHS Foundation Trust, Prestwich, UK
[3]Department of Psychology, School of Human and Health Sciences, University of Huddersfield, Huddersfield, UK
[4]North West Boroughs Healthcare NHS Foundation Trust, Warrington, United Kingdom
[5]Lancashire Care NHS Foundation Trust, Preston, UK
[6]Centre for Mental Health and Risk, Division of Psychology and Mental Health, School of Health Sciences, Faculty of Biology, Medicine and Health, University of Manchester, Manchester Academic Health Science Centre, Manchester, UK
[7]Mental Health and Criminal Justice Group, University of Central Lancashire, Preston, UK
[8]Manchester Centre for Health Economics, Division of Population Health, Health Services Research and Primary Care, School of Health Sciences, Faculty of Biology, Medicine and Health, University of Manchester, Manchester Academic Health Science Centre, Manchester, UK
[9]Offender Health Research Network, Division of Psychology and Mental Health, School of Health Sciences, Faculty of Biology, Medicine and Health, University of Manchester. Manchester Academic Health Science Centre, Manchester, UK

**Acknowledgements** We thank the service users, prison staff and healthcare professionals who participated in the focus groups during phase 1 of the study. We thank Fiona Edgar and Tracy Millington, our experts-by-experience, for their valuable insights and contributions and the COVER project steering group for their ongoing input and support. We also thank the prison governor and our local collaborators in the prison for their support. We thank the local Clinical Research Network for their assistance.

**Contributors** KA and TW conceived and designed the study and applied for funding. KG drafted the original protocol. KA, KG and HM led the development of the prison-modified MSC intervention; are responsible for drafting and revising the protocol manuscript; have given final approval for the version to be published and are accountable for all aspects of the work. HM and BJD led the write-up of the protocol manuscript under the supervision of KG. KG is the research project manager and HM/BJD are the research assistants on the study. KA, KG, HM, TW, SR, LR, JS and RM coled the development of COVER and participated in the design of the study. KA and JS provide the senior academic oversight on all aspects of the feasibility study. KG, HM and BJD lead the patient and public involvement (PPI). FE and TM have provided expert by experience input throughout the project. All authors read and approved the final protocol manuscript.

**Funding** This project was funded by the National Institute for Health Research (NIHR) Research for Patient Benefit Programme (reference PB-PG-1013-32075). This article presents independent research funded by the NIHR. The views expressed are those of the authors and not necessarily those of the NHS, the NIHR or the Department of Health and Social Care. The study sponsor is The University of Manchester and the host NHS Trust is Greater Manchester Mental Health NHS Foundation Trust.

**Disclaimer** The views and opinions expressed therein are those of the authors and do not necessarily reflect those of the RfPB programme, NIHR, NHS or the Department of Health.

**Competing interests** None declared.

**Patient consent for publication** Not required.

**Ethics approval** Ethical approval has been granted by the West of Scotland Research Ethics Committee (REC) 3 (reference: 16/WS/0155). Recruitment is currently ongoing; to date, 30 participants have entered the phase 4 trial.

**Provenance and peer review** Not commissioned; externally peer reviewed.

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
