## [Reviewer comments · BMJ Open]

This paper was submitted to a another journal from BMJ but declined for publication following peer review. The authors addressed the reviewers' comments and submitted the revised paper to BMJ Open. The paper was subsequently accepted for publication at BMJ Open.

(This paper received three reviews from its previous journal but only two reviewers agreed to published their review.)

ARTICLE DETAILS

TITLE (PROVISIONAL)	An acceptability and feasibility pilot randomised controlled trial of medical skin camouflage for recovery of women prisoners with self-harm scarring (COVER): the study protocol.
AUTHORS	Mitchell, Heather; Abel, Kathryn; Dunlop, Brendan James; Walker, Tammi; Ranote, Sandeep; Robinson, Louise; Edgar, Fiona; Millington, Tracy; Meacock, Rachel; Shaw, Jennifer; Gutridge, Kerry

VERSION 1 – REVIEW

REVIEWER	Amanda Perry University of York, UK.
REVIEW RETURNED	13-Feb-2018

GENERAL COMMENTS	This is a clearly written protocol which adheres to the SPIRIT checklist. My only minor comment would be to add something about the justification of the proposed sample size why have 40 women? The justification may be difficult to ascertain but some reference to this would be a useful addition. Under the exclusion criteria is it important that the women's first language is English? Should this be a reason to exclude someone? It is not explicit but I assume that the training for the 6-10 women will be conducted in one group? What about considering the cost of treatment versus usual care? You state that MSC is not readily available why is this? How will something like this be sustained going forwards?
--

REVIEWER	Annette T Maruca University of Connecticut, US
REVIEW RETURNED	29-Apr-2018

GENERAL COMMENTS	The significance of self harm scarring was not well documented by the authors. There is very little information on the importance of scarring for incarcerated women to support the extensive study protocol. As a researcher in corrections, this topic area would be of interest if the authors showed evidence of its significance in the introduction that is very brief. Is there evidence that scarring affects reentry to the community? if so how? There are sections in the manuscript that suggest phases 1 and 2 (focus groups with women) has been completed noted on pages 6, 7, 18 and 21. Results of these phases would have provided readers
---

	with a greater understanding of the need for the MSC intervention and hear from the voices of the incarcerated women themselves on their view this intervention. Then lead into the next steps of the project. Each phase should inform then next phase. There were just brief mentions of some formative changes based on feed back from the women. Where are the themes from data analysis? There were parts that were too confusing especially trying to remember the aim(s) of the study since each phase is unique. My recommendation is to report the results of phase 1 and 2 rather than the protocol study. These results would advance knowledge of this topic more than the protocol
--	---

VERSION 1 – AUTHOR RESPONSE

Reviewer 1:

5) My only minor comment would be to add something about the justification of the proposed sample size why have 40 women?

Response: The justification for the sample size has been added to the text.

6) Under the exclusion criteria is it important that the women's first language is English? Should this be a reason to exclude someone?

Response: The exclusion criteria do not specify that a participant's first language needs to be English. We specify that they need to be able to provide informed consent. Women in the study need to be able to understand spoken English so they can work with their peers who will be providing the intervention.

7) It is not explicit but I assume that the training for the 6-10 women will be conducted in one group?

Response: We have made it explicit that it is a group training session.

8) What about considering the cost of treatment versus usual care?

Response: Resource use collection is being piloted to see if it is possible to collect all the information needed to calculate the cost of the intervention versus usual treatment in a large scale trial. This has been made explicit in the text.

9) You state that MSC is not readily available why is this? How will something like this be sustained going forwards?

Response: These are both important questions which will be considered during the research and reported with the results, rather than in the protocol.

Reviewer 2:

10) The significance of self harm scarring was not well documented by the authors. There is very little information on the importance of scarring for incarcerated women to support the extensive study protocol. As a researcher in corrections, this topic area would be of interest if the authors showed evidence of its significance in the introduction that is very brief. Is there evidence that scarring affects reentry to the community? if so how?

Response: There is very little published information on this topic. The significance of self-harm scarring will be explored in the qualitative aspects of the research and reported in papers which focus on the results of the research.

11) There are sections in the manuscript that suggest phases 1 and 2 (focus groups with women) has been completed noted on pages 6, 7, 18 and 21. Results of these phases would have provided readers with a greater understanding of the need for the MSC intervention and hear from the voices of the incarcerated women themselves on their view this intervention. Then lead into the next steps of the project. Each phase should inform then next phase. There were just brief mentions of some formative changes based on feed back from the women. Where are the themes from data analysis?

Response: This paper is designed to provide a protocol for delivery and testing of a novel intervention within women's prison. The focus group analysis will be reported separately as it is an outcome of the research.

VERSION 2 – REVIEW

REVIEWER	Amanda Perry University of York UK
REVIEW RETURNED	19-Jun-2018
GENERAL COMMENTS	This revision address all previous comments about the protocol. I am satisfied that this protocol is ready for publication.